# Sex- and age- specific normal values of left ventricular functional and myocardial mass parameters using threshold-based trabeculae quantification

Zsófia Gregor[1], Anna Réka Kiss[1], Liliána Erzsébet Szabó[1], Attila Tóth[1], Kinga Grebur[1], Márton Horváth[1], Zsófia Dohy[1], Béla Merkely[1,2], Hajnalka Vágó[1,2], Andrea Szűcs[1]*

1 Heart and Vascular Center of Semmelweis University, Budapest, Hungary, 2 Department of Sports Medicine, Semmelweis University, Budapest, Hungary

* szucsand@gmail.com

**Data Availability Statement:** All relevant data are within the paper.

## Abstract

### Background

The threshold-based (TB) trabeculated and papillary muscle mass (TPM) quantification method for cardiac MRI (CMR) calculates different values than conventional contouring techniques. We aimed to identify the sex- and age-related normal reference ranges for left ventricular (LV) myocardial mass values, volumetric and functional parameters and the correspondence of these parameters using the TB method.

### Methods

Healthy European adults (n = 200, age: 39.4 ± 12 years, males: 100) were examined with CMR and evaluated with a TB postprocessing method. They were stratified by sex and age (Group A: 18–29, Group B: 30–39, Group C: 40–49, Group D: >50 years). The calculated parameters were indexed to body surface area ($i$).

### Results

The normal reference ranges for the studied parameters were assessed in each age group. Significant biometric differences in LV parameters and mass-to-volume ratios were found between males and females, and the left ventricular compacted myocardial mass (LVCM$i$) and TPM$i$ differences remained significant after stratification by age. Unlike other LV volumetric and functional parameters and mass-to-volume ratios, the TPM$i$, the LVCM$i$ and the TPM$i$-to-LVCM$i$ ratio did not differ among age groups in males or females. This finding was strengthened by the lack of correlation between TPM$i$ and age.

### Conclusions

Age- and sex-related normal reference ranges for LV volumetric and functional parameters and LVCM$i$ and TPM$i$ values were established using a TB postprocessing method. TPM$i$, LVCM$i$ and their ratio did not change over time. The TPM$i$-to-LVCM$i$ and the mass-to-

**Funding:** Supported by the ÚNKP-19-3-II New National Excellence Program of the Ministry for Innovation and Technology. http://www.unkp.gov.hu This study was supported by the National Research, Development and Innovation Office of Hungary (NKFIA; NVKP_16-1-2016-0017 National Heart Program). https://nkfih.gov.hu The research was financed by the Thematic Excellence Program (Tématerületi Kiválósági Program 2020-4.1.1-TKP2020) of the Ministry for Innovation and Technology in Hungary within the framework of the Therapeutic Development and Bioimaging programs of the Semmelweis University. https://nkfih.gov.hu/english/nrdi-fund/thematic-excellence-programme-2020-411-tkp2020 The funders had no role in study design, data collection and analysis, decision to publish, or preparation of the manuscript.

**Competing interests:** The authors have declared that no competing interests exist.

volume ratios might have clinical utility in the differential diagnosis of conditions with LV hypertrabeculation.

## Introduction

Several studies have been published about left ventricular (LV) functional parameters in both pathological and physiological conditions [1–3]. The border between normal and excessive LV trabeculation is undefined [2,4,5]; however, normal values of the trabecular mass might assist with the diagnosis of pathological conditions with hypertrabeculation, e.g., left ventricular noncompaction (LVNC), hypertrophic cardiomyopathy (HCM), dilated cardiomyopathy (DCM), and congenital heart diseases.

A number of studies have already been performed in various populations using different methods and vendors since trabeculae quantification became available [6–8].

However, the established values of LV trabeculation are not comparable due to the use of different techniques, the lack of intervendor agreements, and the heterogeneous study populations.

The threshold-based (TB) trabeculated and papillary muscle (TPM) quantification method is based on the differing signal intensities of blood and myocardium and has been proven to be an excellent evaluation method with better interobserver agreement than conventional techniques [9–11]. The determined volumes are lower and the ejection fraction (EF) and myocardial mass values are higher when using this method than when using standard contouring techniques based on endocardial and epicardial contours [12]. Therefore, novel reference ranges are required.

To the best of our knowledge, no studies have been performed to determine the normal reference ranges for LV functional parameters and myocardial mass values using the TB method. In addition, there are no data about sex- and age-specific normal values of the LV trabecular mass.

We aimed to describe the normal reference ranges for LV functional parameters, compacted myocardial mass (LVCM), TPM and the correspondences of these parameters based on a healthy European population divided by sex and age using the TB method.

## Materials and methods

### Study population

Two hundred healthy adult European volunteers were enrolled in this single-center study (mean age: 39.4 ± 12 years, males: 100, mean EF: 68.7 ± 5.1%). Each participant completed a questionnaire about demographic characteristics, cardiovascular symptoms, medical history, medication and sport activity. The following exclusion criteria were applied: presence of any congenital cardiac abnormalities or acquired ischemic heart diseases, arrhythmias, valvular heart diseases, cardiomyopathies, other cardiac diseases or sudden cardiac death in the family history. Furthermore, participants with extracardiac disorders, including hypertension-related, pulmonary, nephrology, gastrointestinal, metabolic, autoimmune, hormonal, psychiatric, oncologic or neuromuscular diseases or other hereditary conditions, were excluded. None of the participants had received medical therapy. Athletes with competitive sport activity (>6 hours/week) were also ruled out [13].

In addition to cardiac MRI (CMR) examinations, blood pressure measurements and 12-lead resting electrocardiography were performed for each participant.

**Table 1. Baseline characteristics of the total population and the subgroups.**

| | | Age (years) | Weight (kg) | Height (m) | BMI (kg/m$^2$) | BSA (m$^2$) |
|---|---|---|---|---|---|---|
| **Total population** (n = 200) | Total | 39.4 ± 12.0 | 74.4 ± 15.0 | 1.7 ± 0.1 | 24.3 ± 3.6 | 1.9 ± 0.2 |
| | Male | 39.6 ± 12.3 | 85.0 ± 11.1 | 1.8 ± 0.1 | 25.8 ± 3.1 | 2.1 ± 0.2 |
| | Female | 39.2 ± 11.8 | 63.9 ± 10.1 | 1.7 ± 0.1 | 22.8 ± 3.3 | 1.7 ± 0.1 |
| | **p** | **0.82** | **<0.0001** * | **<0.0001** * | **<0.0001** * | **<0.0001** * |
| **Group A** (n = 50) | Male | 24.5 ± 3.2 | 80.2 ± 10.6 | 1.8 ± 0.1 | 24.2 ± 3.4 | 2.0 ± 0.1 |
| | Female | 24.1 ± 3.2 | 60.3 ± 8.2 | 1.7 ± 0.1 | 21.1 ± 2.9 | 1.7 ± 0.1 |
| | **p** | **0.66** | **<0.0001** * | **<0.0001** * | **0.0012** * | **<0.0001** * |
| **Group B** (n = 50) | Male | 33.6 ± 2.6 | 87.1 ± 12.8 | 1.8 ± 0.1 | 26.4 ± 2.8 | 2.1 ± 0.2 |
| | Female | 33.6 ± 2.7 | 64.4 ± 13.4 | 1.7 ± 0.1 | 22.6 ± 4.1 | 1.7 ± 0.2 |
| | **p** | **0.96** | **<0.0001** * | **<0.0001** * | **0.0004** * | **<0.0001** * |
| **Group C** (n = 50) | Male | 44.8 ± 3.1 | 87.5 ± 12.1 | 1.8 ± 0.1 | 26.4 ± 3.0 | 2.1 ± 0.2 |
| | Female | 44.8 ± 2.3 | 63.5 ± 8.7 | 1.7 ± 0.1 | 22.7 ± 2.5 | 1.7 ± 0.1 |
| | **p** | **0.92** | **<0.0001** * | **<0.0001** * | **<0.0001** * | **<0.0001** * |
| **Group D** (n = 50) | Male | 55.7 ± 4.3 | 85.1 ± 7.3 | 1.8 ± 0.1 | 26.4 ± 2.8 | 2.1 ± 0.1 |
| | Female | 54.4 ± 3.4 | 67.2 ± 8.7 | 1.7 ± 0.04 | 24.6 ± 2.8 | 1.8 ± 0.1 |
| | **p** | **0.25** | **<0.0001** * | **<0.0001** * | **0.0289** * | **<0.0001** * |

BMI: Body mass index, BSA: Body surface area.

Group A: 20–29 years, Group B: 30–39 years, Group C: 40–49 years, Group D: ≥ 50 years.

*p< 0.05.

The study population was divided by age as follows: 20–29 years (Group A, n = 50), 30–39 years (Group B, n = 50), 40–49 years (Group C, n = 50) and ≥ 50 years (Group D, n = 50), and each subgroup contained an equal number of male and female participants. The baseline characteristics are seen in Table 1.

All procedures performed in this study were in accordance with the 1964 Helsinki declaration and its later amendments or comparable ethical standards. Ethical approval was obtained from the Central Ethics Committee of Hungary, and all participants provided written informed consent.

## Image acquisition and analysis

CMR examinations were performed on 1.5 T MRI scanners (Achieva, Philips Medical System and MAGNETOM Aera, Siemens Healthineers).

Retrospectively gated, balanced steady-state free precession (bSSFP) cine images were acquired in conventional two-chamber, three-chamber and four-chamber long-axis views. Breathhold short-axis cine images from base to apex were obtained. Contrast agent was not administered.

Medis Suite software was used for the postprocessing analysis (Medis Suite, version 3.2, Medis Medical Imaging Systems).

The LV parameters were calculated by a TB algorithm of the software (MassK module of the Medis Suite program). The method is based on the differential signal intensity of blood and myocardium. The program identifies each voxel within the epicardial contour as either blood or myocardium according to the chosen threshold, which was set to the default (50%). Endocardial contours include LV trabeculation and papillary muscle; therefore, the voxels detected as myocardium within the endocardial borders are identified as trabeculation (Fig 1). Unlike conventional postprocessing techniques based on manually contoured endocardial and

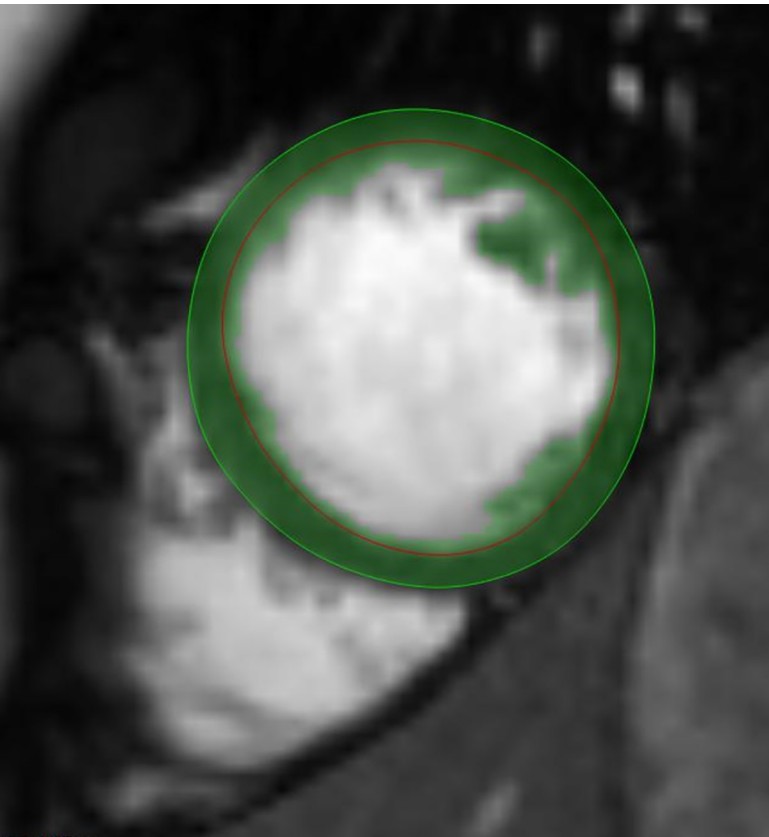

**Fig 1. The threshold-based (TB) method.** The program identifies each voxel within the left ventricular epicardial contour (green line) as either blood or myocardium according to the chosen threshold. Inside the endocardial contour (red line) the green area represents the trabeculated and papillary muscle mass (TPMi).

epicardial borders [9], the TB semiautomatic quantification method is highly reproducible and is independent of the experience of the observer. The details of this technique and the above mentioned advantages were emphasized in several studies [9–11,14].

The end-diastolic and end-systolic phases were identified, and semiautomatic tracing of the epicardial and endocardial contours was corrected manually in all slices from base to apex. The contours were made by two observers (ZG with 4 years of experience, and ARK with 5 years of experience) with excellent interobserver variability, and it was determined by the global intraclass correlation coefficient, which was 0.92 (interpreted as follows: 0.4–0.75 = fair to good, and greater than 0.75 = excellent).

Short-axis images were applied to calculate the following LV parameters using the TB method: end-diastolic volume (EDV), end-systolic volume (ESV), stroke volume (SV), cardiac output (CO), EF, end-diastolic total myocardial mass and end-diastolic TPM. In our study, LVCM was calculated as the difference between LV end-diastolic total muscle mass and TPM. In the following, we used only LVCM and TPM for the characterization of the myocardial mass.

All of the measured parameters were indexed to body surface area (*i*). The trabeculated and papillary muscle mass-to-myocardial mass (TPM*i*/LVCM*i*), the myocardial mass-to-end-diastolic volume (LVCM*i*/EDV*i*) and the trabeculated and papillary muscle mass-to-end-diastolic volume (TPM*i*/EDV*i*) ratios were created to assess the correspondence of the LV parameters.

## Statistical analysis

The Kolmogorov-Smirnov test was used to assess data distribution. Continuous variables were reported either as the mean ± standard deviation (SD) or the median [interquartile range] as appropriate. The 95% confidence intervals (CIs) were calculated to assess the normal range for LV parameters. Independent-sample t-tests were applied to compare parameters that fit a normal distribution; otherwise, the Mann-Whitney test was used. The comparison of the functional parameters of the different age groups was performed with one-way analysis of variance (ANOVA) with the Tukey–Kramer post hoc test. Linear correlations were assessed using the Pearson correlation coefficient. A p value < 0.05 was considered statistically significant. The intraclass correlation coefficients (ICCs) were calculated to assess interobserver variability. The statistical analysis was performed using MedCalc Statistical Software version 17.9.5. (MedCalc Software).

# Results

## Sex- and age-specific normal values of LV parameters

The normal reference ranges of the LV functional parameters, the LVCM, the TPM and the derived parameters were assessed in the total study population and in each age group divided by sex (Table 2).

Regarding the total population, the following functional parameters differed significantly between sexes: the EDV*i*, ESV*i*, LVCM*i* and TPM*i* were higher and the EF was lower in males than in females.

After dividing the participants by age and sex, unlike the functional parameters, the LVCM*i* and TPM*i* values differed significantly in all age groups; however, the difference in the TPM*i*/LVCM*i* ratio remained nonsignificant between sexes (Table 2). The LVCM*i*/EDV*i* and TPM*i*/EDV*i* ratios also showed significant differences between sexes in the age groups (Table 2).

## Changes of the LV parameters with age

In the analysis of age-related changes in LV functional parameters, the EDV*i*, the ESV*i*, the SV*i* and the CO*i* decreased with age: Group A had the highest and Group D had the lowest values in both sexes (Table 2). Although no significant differences were found between the groups, the EF showed a weak positive correlation with age (r = 0.14, p = 0.04).

Regarding the changes in myocardial masses, the TPM*i*, the LVCM*i*, and their ratios did not differ among age groups in males or females. However, due to the changes in volumetric parameters, LVCM*i*/EDV*i* and TPM*i*/EDV*i* were also altered among the age groups, showing an increase in both sexes (Table 2).

## Correlations between TPM and phenotype characteristics

According to an analysis of the relationship between TPM*i* and the phenotypic characteristics of the total population, with the exception of age and CO*i*, all of the observed parameters correlated with TPM*i*: the strongest relationship was with LVCM*i*, which was followed by BSA, ESV*i* and EDV*i* (Fig 2).

After dividing the population by sex and age, these significant differences mostly disappeared (Table 3).

Notably, TPM*i* and EF had a negative correlation independent of sex in all age groups.

# Discussion

In this study, we established age- and sex-related normal reference ranges for LV functional parameters and LVCM*i* and TPM*i* values using the TB method in a healthy European cohort.

**Table 2. Reference ranges of age and sex related left ventricular parameters and changes with the age.**

| | Group | | Male | Female | p (Male vs Female) |
|---|---|---|---|---|---|
| **EDVi (ml/m$^2$)** | Total | Mean ± SD | 69.5 ± 10.7 | 64.5 ± 8.5 | **0.0003**[*] |
| | | 95% CI | 51.8–93.8 | 50.0–82.2 | |
| | A | Mean ± SD | 73.2 ± 11.5[#] | 68.3 ± 10.4[#] | **0.12** |
| | | 95% CI | 54.5–96.9 | 53.9–97.2 | |
| | B | Mean ± SD | 68.6 ± 10.4 | 64.7 ± 6.1 | **0.11** |
| | | 95% CI | 48.7–93.2 | 53.0–77.6 | |
| | C | Mean ± SD | 71.4 ± 9.5 | 66.0 ± 7.5[#] | **0.028** [*] |
| | | 95% CI | 53.8–91.3 | 51.5–81.5 | |
| | D | Mean ± SD | 64.6 ± 10.1[¤] | 58.9 ± 6.8[¤ $] | **0.023** [*] |
| | | 95% CI | 51.8–87.0 | 48.0–71.6 | |
| | **p (age groups)** | | **0.027** [*] | **0.001** [*] | |
| **ESVi (ml/m$^2$)** | Total | Mean ± SD | 22.7 ± 5.2 | 19.3 ± 4.1 | **<0.0001**[*] |
| | | 95% CI | 13.7–33.6 | 12.3–29.7 | |
| | A | Mean ± SD | 25.1 ± 5.3[#] | 20.7 ± 4.1[#] | **0.002** [*] |
| | | 95% CI | 17.1–36.0 | 14.9–30.0 | |
| | B | Mean ± SD | 22.6 ± 4.7 | 19.9. ± 3.7[#] | **0.032** [*] |
| | | 95% CI | 13.7–32.4 | 13.8–27.8 | |
| | C | Mean ± SD | 22.2 ± 4.9 | 19.7 ± 4.5[#] | **0.07** |
| | | 95% CI | 14.0–33.6 | 12.1–29.5 | |
| | D | Mean ± SD | 21.1 ± 5.4[¤] | 16.8 ± 3.0[¤ & $] | **0.001** [*] |
| | | 95% CI | 12.2–33.0 | 12.0–22.4 | |
| | **p (age groups)** | | **0.04** [*] | **0.003** [*] | |
| **SVi (ml/m$^2$)** | Total | Mean ± SD | 46.6 ± 7.8 | 45.1 ± 6.1 | **0.13** |
| | | 95% CI | 34.1–61.7 | 34.4–57.9 | |
| | A | Mean ± SD | 48.0 ± 8.3 | 47.4 ± 7.1[#] | **0.77** |
| | | 95% CI | 34.9–66.7 | 38.1–65.9 | |
| | B | Mean ± SD | 46.1 ± 7.1 | 44.7 ± 3.9 | **0.41** |
| | | 95% CI | 34.4–61.0 | 38.1–52.6 | |
| | C | Mean ± SD | 48.8 ± 6.7 | 46.3 ± 5.7 | **0.16** |
| | | 95% CI | 34.6–59.7 | 35.5–56.4 | |
| | D | Mean ± SD | 43.6 ± 8.2 | 42.1 ± 6.3[¤] | **0.49** |
| | | 95% CI | 32.8–61.1 | 31.8–56.1 | |
| | **p (age groups)** | | **0.08** | **0.014** [*] | |
| **COi (l/m$^2$*min)** | Total | Mean ± SD | 3.2 ± 0.8 | 3.0 [2.6,3.6] | **0.56** |
| | | 95% CI | 1.8–4.7 | 2.2–4.5 | |
| | A | Mean ± SD | 3.6 ± 0.7[#] | 3.3 ± 0.7 | **0.22** |
| | | 95% CI | 2.4–4.7 | 2.4–4.3 | |
| | B | Mean ± SD | 3.2 ± 0.8 | 3.0 ± 0.4 | **0.18** |
| | | 95% CI | 2.1–4.6 | 2.3–3.7 | |
| | C | Mean ± SD | 3.3 ± 0.8 | 3.4 ± 0.8[#] | **0.60** |
| | | 95% CI | 1.8–5.0 | 2.2–5.0 | |
| | D | Mean ± SD | 2.8 ± 0.8[¤] | 2.9 ± 0.5[$] | **0.57** |
| | | 95% CI | 1.0–4.4 | 2.2–4.1 | |
| | **p (age groups)** | | **0.005** [*] | **0.011** [*] | |

(*Continued*)

**Table 2.** (Continued)

| | Group | | Male | Female | p (Male vs Female) |
|---|---|---|---|---|---|
| **EF (%)** | Total | Mean ± SD | 67.2 ± 5.4 | 70.2 ± 4.3 | <**0.0001** * |
| | | 95% CI | 55.6–77.3 | 61.7–79.7 | |
| | A | Mean ± SD | 65.7 ± 4.6 | 69.8 ± 2.3 | **0.0002** * |
| | | 95% CI | 56.1–75.5 | 65.6–73.5 | |
| | B | Mean ± SD | 67.3 ± 4.0 | 69.3 ± 3.9 | **0.07** |
| | | 95% CI | 61.9–74.3 | 61.3–76.4 | |
| | C | Mean ± SD | 68.5 ± 5.9 | 70.3 ± 5.2 | **0.28** |
| | | 95% CI | 53.1–79.6 | 58.4–81.0 | |
| | D | Mean ± SD | 67.4 ± 6.6 | 71.4 ± 4.9 | **0.02** * |
| | | 95% CI | 54.2–79.3 | 63.2–80.1 | |
| | **p (age groups)** | | **0.32** | **0.37** | |
| Total LV Massi (g/m2) | Total | Mean ± SD | 73.7 ± 9.2 | 58.8 ± 7.0 | <**0.0001** * |
| | | 95% CI | 56.3–91.9 | 47.0–73.6 | |
| | A | Mean ± SD | 73.8 ± 8.7 | 59.0 ± 7.4 | <**0.0001** * |
| | | 95% CI | 54.4–89.9 | 48.6–73.6 | |
| | B | Mean ± SD | 72.6 ± 9.1 | 58.1 ± 6.6 | <**0.0001** * |
| | | 95% CI | 56.7–87.3 | 47.2–71.1 | |
| | C | Mean ± SD | 72.7 [69.7,86.4] | 58.8 ± 7.7 | <**0.0001** * |
| | | 95% CI | 66.6–92.5 | 46.8–76.5 | |
| | D | Mean ± SD | 71.4 ± 9.7 | 59.6 ± 6.5 | <**0.0001** * |
| | | 95% CI | 55.0–91.7 | 44.9–70.6 | |
| | **p (age groups)** | | **0.25** | **0.91** | |
| **LVCMi (g/m$^2$)** | Total | Mean ± SD | 50.7 ± 6.9 | 40.7 ± 5.6 | <**0.0001** * |
| | | 95% CI | 38.2–65.9 | 32.1–52.1 | |
| | A | Mean ± SD | 51.4 ± 6.4 | 40.5 [36.6,44.4] | <**0.0001** * |
| | | 95% CI | 37.6–65.7 | 33.3–55.0 | |
| | B | Mean ± SD | 49.7 ± 7.2 | 39.4 ± 5.2 | <**0.0001** * |
| | | 95% CI | 36.8–64.5 | 30.0–50.8 | |
| | C | Mean ± SD | 53.2 ± 6.6 | 41.0 ± 6.4 | <**0.0001** * |
| | | 95% CI | 43.6–67.2 | 31.6–55.7 | |
| | D | Mean ± SD | 48.3 ± 6.7 | 41.0 ± 4.7 | <**0.0001** * |
| | | 95% CI | 39.3–65.1 | 33.4–47.9 | |
| | **p (age groups)** | | **0.07** | **0.61** | |
| **TPMi (g/m$^2$)** | Total | Mean ± SD | 23.0 ± 4.7 | 18.2 ± 3.1 | <**0.0001** * |
| | | 95% CI | 14.7–35.1 | 12.8–24.0 | |
| | A | Mean ± SD | 22.4 ± 3.8 | 17.7 ± 2.8 | <**0.0001** * |
| | | 95% CI | 16.5–31.1 | 12.9–22.7 | |
| | B | Mean ± SD | 22.9 ± 4.4 | 18.8 ± 3.2 | **0.0004** * |
| | | 95% CI | 14.0–34.0 | 13.1–25.5 | |
| | C | Mean ± SD | 23.7 ± 5.0 | 17.8 ± 2.3 | <**0.0001** * |
| | | 95% CI | 16.0–36.1 | 13.1–21.1 | |
| | D | Mean ± SD | 23.1 ± 5.5 | 18.3 ± 3.9 | **0.0008** * |
| | | 95% CI | 11.9–35.7 | 10.5–27.2 | |
| | **p (age groups)** | | **0.82** | **0.59** | |

(*Continued*)

**Table 2.** (Continued)

| | Group | | Male | Female | p (Male vs Female) |
|---|---|---|---|---|---|
| **TPMi/LVCMi (%)** | Total | Mean ± SD | 46.0 ± 9.8 | 45.4 ± 8.8 | **0.66** |
| | | 95% CI | 28.6–64.2 | 30.1–66.2 | |
| | A | Mean ± SD | 43.9 ± 6.9 | 43.5 ± 7.0 | **0.85** |
| | | 95% CI | 33.1–58.0 | 30.1–55.4 | |
| | B | Mean ± SD | 46.8 ± 10.1 | 48.3 ± 9.2 | **0.57** |
| | | 95% CI | 29.2–71.9 | 30.1–65.9 | |
| | C | Mean ± SD | 44.9 ± 9.6 | 44.5 ± 6.9 | **0.86** |
| | | 95% CI | 28.3–63.8 | 32.0–56.5 | |
| | D | Mean ± SD | 48.3 ± 12.0 | 45.2 ± 11.2 | **0.35** |
| | | 95% CI | 27.2–77.1 | 30.2–72.0 | |
| | **p (age groups)** | | **0.39** | **0.24** | |
| **LVCMi/EDVi (g/ml)** | Total | Mean ± SD | 0.74 ± 0.1 | 0.64 ± 0.09 | **<0.0001** * |
| | | 95% CI | 0.56–0.96 | 0.49–0.85 | |
| | A | Mean ± SD | 0.71 ± 0.09 | 0.61 ± 0.07[#] | **<0.0001** * |
| | | 95% CI | 0.56–0.88 | 0.45–0.71 | |
| | B | Mean ± SD | 0.74 ± 0.13 | 0.61 ± 0.07[#] | **0.0001** * |
| | | 95% CI | 0.54–1.12 | 0.49–0.74 | |
| | C | Mean ± SD | 0.75 ± 0.09 | 0.62 ± 0.08[#] | **<0.0001** * |
| | | 95% CI | 0.57–0.94 | 0.48–0.77 | |
| | D | Mean ± SD | 0.75 ± 0.09 | 0.70 ± 0.10[¤ & $] | **0.07** |
| | | 95% CI | 0.60–0.99 | 0.52–0.87 | |
| | **p (age groups)** | | **0.43** | **<0.001** * | |
| **TPMi/EDVi (g/ml)** | Total | Mean ± SD | 0.32 [0.28,0.39] | 0.28 [0.25,0.31] | **<0.0001** * |
| | | 95% CI | 0.22–0.50 | 0.20–0.42 | |
| | A | Mean ± SD | 0.31 ± 0.06 | 0.26 ± 0.04[#] | **0.0035** * |
| | | 95% CI | 0.22–0.47 | 0.18–0.36 | |
| | B | Mean ± SD | 0.32 [0.28,0.38] | 0.29 ± 0.05 | **0.0145** * |
| | | 95% CI | 0.24–0.49 | 0.19–0.39 | |
| | C | Mean ± SD | 0.34 ± 0.08 | 0.26[0.25,0.28][#] | **0.0017** * |
| | | 95% CI | 0.22–0.52 | 0.22–0.39 | |
| | D | Mean ± SD | 0.36 ± 0.08 | 0.32 ± 0.08[¤ $] | **0.05** |
| | | 95% CI | 0.22–0.59 | 0.20–0.47 | |
| | **p (age groups)** | | **0.14** | **0.006** * | |

Values presented in either mean ± SD for normally distributed data or median [interquartile range] for non-normally distributed data. The 95% CI was calculated for reference ranges.

*p< 0.05.

EDVi: End-diastolic volume index, ESVi: End-systolic volume index, SVi: Stroke volume index, COi: Cardiac output index, EF: Ejection fraction, LVCMi: LV end-diastolic compacted myocardial mass index, TPMi: LV end-diastolic papillary and trabeculated muscle mass index.

SD: Standard deviation, CI: Confidence interval, LV: Left ventricular.

¤ p < 0.05 vs Group A

& p < 0.05 vs Group B

$ p < 0.05 vs Group C

# p < 0.05 vs Group D.

However, as trabeculated and papillary muscle mass is included in the endocardial contours but added to the myocardial mass instead of the volumetric values, the LV volumes and LV mass values differ significantly compared to those calculated with traditional techniques. Thus,

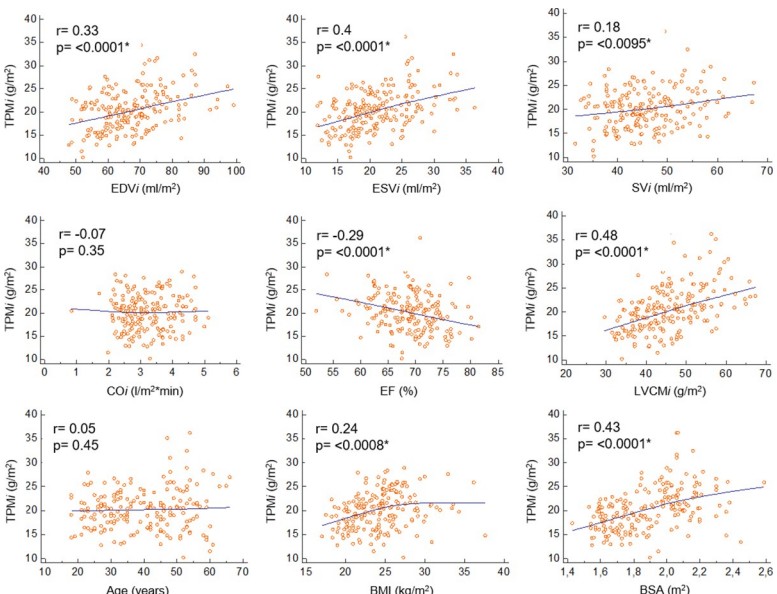

**Fig 2. Correlations with the TPMi in the total population.** EDVi: End-diastolic volume index, ESVi: End-systolic volume index, SVi: Stroke volume index, COi: Cardiac output index, EF: Ejection fraction, LVCMi: LV end-diastolic compacted myocardial mass index, TPMi: LV end-diastolic papillary and trabeculated muscle mass index, BMI: Body mass index, BSA: Body surface area. *p< 0.05.

**Table 3. Correlations with the TPMi divided by age.**

| | | TPMi | | | | | | | |
|---|---|---|---|---|---|---|---|---|---|
| | | Group A | | Group B | | Group C | | Group D | |
| | | r | p | r | p | r | p | r | p |
| **EDVi** (ml/m²) | Male | 0.28 | 0.18 | 0.41 | 0.04 * | 0.05 | 0.81 | 0.47 | 0.02 * |
| | Female | 0.35 | 0.09 | 0.21 | 0.32 | 0.36 | 0.08 | 0.04 | 0.84 |
| **ESVi** (ml/m²) | Male | 0.36 | 0.08 | 0.35 | 0.08 | 0.27 | 0.2 | 0.46 | 0.02 * |
| | Female | 0.18 | 0.4 | 0.2 | 0.34 | 0.52 | 0.007 * | 0.06 | 0.78 |
| **SVi** (ml/m²) | Male | 0.15 | 0.47 | 0.37 | 0.07 | -0.08 | 0.7 | 0.28 | 0.18 |
| | Female | 0.37 | 0.07 | 0.13 | 0.53 | 0.05 | 0.8 | 0.02 | 0.93 |
| **COi** (l/m2*min) | Male | -0.11 | 0.59 | -0.04 | 0.85 | -0.4 | 0.05 | 0.16 | 0.44 |
| | Female | 0.01 | 0.94 | -0.02 | 0.91 | 0.04 | 0.86 | -0.34 | 0.1 |
| **EF** (%) | Male | -0.26 | 0.2 | -0.12 | 0.55 | -0.22 | 0.29 | -0.21 | 0.3 |
| | Female | -0.05 | 0.8 | -0.15 | 0.48 | -0.45 | 0.02 * | -0.002 | 0.99 |
| **LVCMi** (g/m²) | Male | 0.39 | 0.05 | 0.18 | 0.39 | -0.2 | 0.34 | 0.26 | 0.20 |
| | Female | 0.35 | 0.09 | 0.21 | 0.32 | 0.43 | 0.03 * | 0.03 | 0.90 |
| **BMI** (kg/m²) | Male | 0.007 | 0.97 | -0.3 | 0.14 | 0.06 | 0.78 | -0.19 | 0.37 |
| | Female | 0.33 | 0.11 | -0.08 | 0.7 | 0.01 | 0.96 | 0.26 | 0.22 |
| **BSA** (m²) | Male | 0.11 | 0.59 | -0.26 | 0.21 | 0.06 | 0.78 | 0.04 | 0.85 |
| | Female | 0.35 | 0.09 | -0.04 | 0.83 | 0.28 | 0.18 | 0.1 | 0.63 |
| **Age** (year) | Male | -0.16 | 0.44 | 0.07 | 0.75 | 0.18 | 0.40 | -0.007 | 0.98 |
| | Female | 0.19 | 0.37 | -0.07 | 0.75 | 0.09 | 0.67 | -0.08 | 0.69 |

EDVi: End-diastolic volume index, ESVi: End-systolic volume index, SVi: Stroke volume index, COi: Cardiac output index, EF: Ejection fraction, LVCMi: LV end-diastolic compacted myocardial mass index, TPMi: LV end-diastolic papillary and trabeculated muscle mass index, BMI: Body mass index, BSA: Body surface area.
*p< 0.05.

novel normal ranges should be established, including reference ranges for LV trabeculation specific to both age and sex [9]. The accurate determination of the trabecular mass values is clinically relevant in conditions presenting with physiological and pathological LV hypertrabeculation [15–17].

### Sex- and age-specific normal values

Regarding the total population, the biometric differences between sexes were in accordance with previously published data, namely, males had significantly higher volumes and myocardial mass values and lower EF than females, regardless of the use of different postprocessing techniques [3,18,19]. After stratifying participants by age, the LVCM*i* and TPM*i* values differed significantly between sexes; however, some of the functional parameters did not show these significances because of the small numbers in the subgroups.

Previous studies described age-related changes in LV functional parameters, but we did not find information about sex-related differences based on age [1,20–23].

For trabeculae quantification, Fernandez-Golfin and Bentatou described trabecular masses based on a conventional method in a diverse study population with healthy subjects, patients with valvular disorders and patients with cardiomyopathy and in a healthy population, but the groups were not divided by age [6,24]. Trabeculated myocardium volumes and percentiles have also been measured in a healthy population and in patients with hypertension, ischemic heart disease and different cardiomyopathies, but these studies did not stratify by age either [8,25]. Andre et al. and Cai et al. described the LV trabeculation of healthy subjects in different age groups but with differing techniques: the former group described the trabecular volume, while the latter group used fractal analysis [7,26].

To the best of our knowledge, there are no publications focusing on both age- and sex-related changes in LV trabecular mass; thus, we provide age- and sex-specific normal reference ranges for LV TPM.

### Changes with increasing age

Analysis of the age-related changes in LV functional parameters in males and females revealed that all volumetric values decreased with age, and the difference between the youngest and oldest groups was significant. Our results are in line with other echocardiographic and CMR studies, namely, the volumetric parameters decreased in accordance with age [1,27–29].

In our study, EF showed a weak positive correlation with age, which was strengthened by Fiechter et al. and Nikitin et al. [27,28], while other studies revealed that aging did not influence LV function [1,29,30].

Furthermore, we observed that the TPM*i* and LVCM*i* did not change in either males or females over time. We did not find information about age-dependent changes in the TPM*i* calculated with the TB method. Bentatou et al. were the only investigators who described that TPM*i* decreased with increasing age; however, the papillary muscles were included in the compact myocardial mass; this method differs from our technique, which considers papillary muscles in the trabeculated muscle mass [6].

Notably, depending on the analysis software, papillary muscle mass can be counted in either compact or trabeculated muscle mass, which might result in differences between investigators. However, Andre et al. described that this difference is not significant [24].

Opposing results regarding LVCM*i* have been reported: an echocardiographic study revealed an increase in LVCM*i* [31], while other studies conducted with CMR using conventional technique described a slight decrease with increasing age [27,32,33]. Moreover, in contrast to the above-mentioned results, the LVCM*i* did not change significantly with age in

either males or females according to several CMR and echocardiographic investigations [1,21,34]. These results were confirmed with autopsy and are also in accordance with our findings [35]. The underlying pathophysiological mechanism could be the age-related loss of myocytes and compensatory reactive cellular hypertrophy, which maintains the total weight of the myocardium [36].

## Normal values and changes in the ratios

The TPM$i$/LVCM$i$ ratio was not significantly different between sexes, which might imply a sex-independent connection between trabecular mass and LV mass, and no changes were observed among the age groups.

Chuang et al. observed similar values for TPM$i$/LVCM$i$ ratios measured with the conventional contouring technique, but they did not examine this relation in different age groups [8]. Fernandez described smaller ratios in a healthy subgroup with a lower number of patients using another method; however, this group was not divided by sex and age [24]. The normal values of TPMi/LVCMi ratio might have additive value in the diagnosis of different conditions with excessive LV trabeculation.

Regarding the mass/volume ratios, after dividing the total population by sex, the LVCM$i$/EDV$i$ and TPM$i$/EDV$i$ remained significantly different in most age groups, and this difference can be explained by the significantly different myocardial masses in males and females. Both the LVCM$i$/EDV$i$ and TPM$i$/EDV$i$ ratios increased with age due to the age-dependent decrease in volumetric parameters. These results are in accordance with those of other studies describing age-related changes in the LVCM$i$/EDV$i$ ratio [1,31]. We found only one correspondence of age and trabeculation to EDV ratio; in contrast to our results, trabeculae were expressed as volumes, and the ratio had a weak negative correlation with age [8].

Czimbalmos et al. applied the LV mass-to-LV end-diastolic volume ratio to distinguish HCM from an athlete's heart [16]. Thus, these myocardial mass-to-volume ratios could have possible clinical utility in enabling the differential diagnosis of pathological and physiological conditions associated with hypertrophy or hypertrabeculation.

## Correlation with the TPMi

The lack of connection between TPMi and age in the total population corresponds to TPMi remaining unchanged over time after dividing the population by age; however, the underlying cause is still unknown based on other studies.

TPM$i$ and LVCM$i$ showed a strong positive correlation in the total population. Chuang et al. also described the trabeculation and LV mass correlation, although the amount of trabeculation was expressed as volumes [8]. Janik et al. described that papillary and trabecular masses correlate with ventricular mass; however, they observed a diverse population of patients with concentric or eccentric hypertrophy and normal controls, and the participants were not divided by sex or age [15].

In our study, there was a positive correlation between TPM$i$ and BSA in the total population, but no data were found in the literature about this correspondence.

Similar to our study, in Chuang et al., Andre et al. and Bentatou et al.'s studies, LV trabeculae showed a strong relationship with EDV and ESV, measured either by volume or by myocardial mass [6,8,24].

In concordance with our results, the lack of connection between TrM$i$ and BMI was also described previously [33].

After stratifying our total population by age, there was no correlation between TPM$i$ and the other observed parameters, which can be explained by the small number of subgroups.

Interestingly, there was an inverse correlation between TPM$i$ and EF, as in Bentatou's research [6]. The inverse relationship between TPM$i$ and EF might be explained by the association between lower EF and higher myocardial mass (e.g., in males), and LVCM$i$ correlated with TPM$i$ in our study.

## Conclusions

We defined age- and sex-related normal values for LV functional parameters and myocardial and trabecular mass using a TB trabeculated and papillary muscle mass quantification method in a healthy European cohort. The TPM$i$, LVCM$i$ and TPM$i$/LVCM$i$ ratio were independent of sex and age, and this result is strengthened by the correlation between them.

Increasing the number of the subgroups would provide even more accurate results. Overall, normal values would be helpful for the determination of trabecular mass and myocardial mass/volume ratios, which might have clinical utility in the differential diagnosis of physiological and pathological conditions with LV hypertrabeculation.

## Limitations

In the TB method, the papillary muscles are added to the trabeculated myocardial mass because of the nature of this technique. Although this study was performed on a large cohort, after dividing by sex and age, the number of subgroups decreased, which may have affected the statistical findings.

Different vendors and third party software companies may have differences in how they specifically implement the algorithm separating the trabecules and papillary muscles from the blood pool and the myocardium. A simple algorithm will not fit every scanner, and some fine-tuning is necessary for different MR vendors. So the different trabecular measuring techniques are not comparable because their intervendor agreements are not established yet.

Current EF and volume quantification uses a stack of thick short-axis slices, and 8–10 mm is usual for Z-direction spatial resolution. Moreover, trabecules and papillary muscles will not cross the slice in an exactly perpendicular fashion, which creates partial volume effects. Depending on the actual path of the trabecules, this will influence the threshold-based quantification. With the advent of advanced isotropic 4D cine techniques, this drawback will be less important in the future.

## Acknowledgments

We thank the technicians who helped performing the cardiac magnetic resonance imaging examinations.

## Author Contributions

**Conceptualization:** Zsófia Gregor, Anna Réka Kiss, Attila Tóth, Béla Merkely, Hajnalka Vágó, Andrea Szűcs.

**Data curation:** Zsófia Gregor, Anna Réka Kiss, Liliána Erzsébet Szabó, Kinga Grebur, Márton Horváth, Zsófia Dohy.

**Formal analysis:** Zsófia Gregor.

**Funding acquisition:** Béla Merkely, Hajnalka Vágó.

**Investigation:** Zsófia Gregor, Anna Réka Kiss, Liliána Erzsébet Szabó, Attila Tóth, Kinga Grebur, Márton Horváth, Zsófia Dohy, Hajnalka Vágó, Andrea Szűcs.

**Methodology:** Zsófia Gregor, Anna Réka Kiss, Liliána Erzsébet Szabó, Attila Tóth, Kinga Grebur, Márton Horváth, Zsófia Dohy, Andrea Szűcs.

**Resources:** Béla Merkely, Hajnalka Vágó.

**Software:** Attila Tóth.

**Supervision:** Béla Merkely, Hajnalka Vágó, Andrea Szűcs.

**Validation:** Attila Tóth, Andrea Szűcs.

**Visualization:** Zsófia Gregor.

**Writing – original draft:** Zsófia Gregor, Anna Réka Kiss.

**Writing – review & editing:** Andrea Szűcs.

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
