## [Decision Letter · Decision Letter 0]

6 Jul 2021

PONE-D-21-13794

Sex- and age- specific normal values of left ventricular functional and myocardial mass parameters using threshold-based trabeculae quantification

PLOS ONE

Dear Dr. Szűcs,

Thank you for submitting your manuscript to PLOS ONE. After careful consideration, we feel that it has merit but does not fully meet PLOS ONE’s publication criteria as it currently stands. Therefore, we invite you to submit a revised version of the manuscript that addresses the points raised during the review process.

ACADEMIC EDITOR: All issues raised by reviewers are required.

We look forward to receiving your revised manuscript.

Kind regards,

Vincenzo Lionetti, M.D., PhD

Academic Editor

PLOS ONE

Journal Requirements:

3. Please note that outmoded terms and potentially stigmatizing labels should be changed to more current, acceptable terminology. Examples: “Caucasian” should be changed to “white” or “of [Western] European descent” (as appropriate); “cancer victims” should be changed to “patients with cancer.” https://journals.plos.org/plosone/s/submission-guidelines#loc-human-subjects-research

Reviewers' comments:

Reviewer's Responses to Questions

**Comments to the Author**

1. Is the manuscript technically sound, and do the data support the conclusions?

Reviewer #1: Yes

Reviewer #2: Partly

Reviewer #3: Yes

2. Has the statistical analysis been performed appropriately and rigorously? 

Reviewer #1: I Don't Know

Reviewer #2: Yes

Reviewer #3: Yes

3. Have the authors made all data underlying the findings in their manuscript fully available?

Reviewer #1: Yes

Reviewer #2: Yes

Reviewer #3: Yes

4. Is the manuscript presented in an intelligible fashion and written in standard English?

Reviewer #1: Yes

Reviewer #2: Yes

Reviewer #3: Yes

5. Review Comments to the Author

Reviewer #1: Cardiac MRI is nowadays considered the most reproducible diagnostic tool for in vivo investigations of excessive trabeculation phenotypes. This paper raises a quite relevant issue, i.e., the current limits in the knowledge of sex-related (besides age-related) normal reference values to be used in the CMR evaluations of myocardial trabeculation. Assessing the normal variability range in different age and sex groups is an important issue in the imaging field for the correct assessment of cardiac diseases, and this topic has been addressed in recent studies (ex: Sci Rep 10:846,2020; Int J Cardiovasc Imaging 2020, 36:2173).

In previous echocardiographic and CMR studies, these volumetric parameters have been shown to be age-dependent, as correctly acknowledged by the Authors in Discussion section. Unfortunately, comparison with previous studies is not always feasible, because of different imaging technologies and/or of different investigation methods (such as including the papillary muscles in the trabeculated muscle mass as in the present paper or in the compact myocardial mass as in other papers). Riffel et al. have also underlined the impact of different strategies on the assessment of the CMR reference values concluding in their CMR study on 362 healthy volunteer that “Quantitative assessment of LV volumes and mass with inclusion of papillary muscles and trabeculae to myocardial mass resulted in significantly different values” (Clin Res Cardiol 2019;108:411). Some of the Authors of the present (under consideration) paper have previously shown that “The TB algorithm could be a consistent method to assess LV and RV CMR values, and to measure trabeculae and papillary muscles quantitatively in various level of experience in CMR (Int J Cardiovasc Imaging 2018 34:1127).

Few data are available as to sex-related differences, corrected by age. The Authors confirmed that males had significantly higher volumes and myocardial mass values and lower EF than females; they also showed that LVCMi and TPMi values (after stratification by age) differed significantly between sexes.

Controversial results have been reported as to possible aging-related changes in LVCMi; anyhow the results of this study regarding LVCMi such as the lack of connection between TrMi and BMI, are confirmed by other studies (see Ref.s # 1,20,32) and/or by anatomic post-mortem investigations that have been conducted in the past (Ref. # 33), respectively.

Although this paper adds a few data to our knowledge of normal populations values for CMR evaluations, its major limit is due to the small number of the subgroups, as stated by the Authors in the final (Limitations of the study) section. Then further studies are required to confirm their (partially novel) data. Such comment might be included in the Conclusions section.

Reviewer #2: This paper is interesting in trying to define "normality values" by sex and age. But it fails in the method, in my opinion. It uses the tool that already exists in the market and is known (although very imprecise). The problem is the ventricular volumes that do not coincide with those observed so far (they are too small).

MRI has another tool to assess precision in the quantification of ventricular volumes. We can quantify the flow that leaves the left ventricle in each beat in the aortic root. If they are normal "healthy" patients, this flow must be the same as that obtained by volumes. This part is not included in this paper. If authors want to measure it like that and validate their method they must demonstrate that it is the correct one. I think that is the first thing before defining normality values.

Reviewer #3: The Authors measured the LV mass and volumes in 200 healthy caucasian controls, using a threshold-based (TB) method; in particular, they focussed on the trabeculated and papillary muscle mass (TPM) and on the left ventricular compacted myocardial mass (LVCM), which have been overlooked in previous studies. The article is interesting, even though several points should be addressed.

-As far as patient stratification is concerned, the older group (>50 year old people) might not be representative of elderly people (>60 y.o.), who might display a different phenotype and different reference values than middle-aged patients (50-60 y.o.).

-The threshold-based (TB) method relies on signal thresholding between the myocardium and the bloodpool, so that different thresholds produce different results by including/excluding not only any small trabecula but also any blood/myocardiuam partial volume areas. This is apparent even in Figure 1, where some small trabeculae (not highlighted in green) can be hardly distinguished from the bloodpool. Besides intra- and inter-operator variability, the variability of TB measurements should be tested on cineSSFP images acquired with/without Gd, as well as with different spatial and temporal resolutions.

-Total LV myocardial mass should be measured in systole and diastole, as a double-check, also because this TB method should detect all trabeculae (which are usualy missed with manual countouring) and yield very similar total mass values between sytole/diastole.

-Were all the volunteers free from medical therapy? Please specify in the Methods.

-Table 2 and 3 are more or less the same, with only different subgroup analysis (per sex vs. per age group). I would suggest to merge the two tables. Moreover, I would suggest to represent a column/row about total LV mass (which should be the sum of TPM+LVCM).

6. PLOS authors have the option to publish the peer review history of their article (what does this mean?). If published, this will include your full peer review and any attached files.

Reviewer #1: No

Reviewer #2: No

Reviewer #3: No

---

## [Author Response · Author response to Decision Letter 0]

12 Aug 2021

Responses to the Reviewers

Reviewer #1 Cardiac MRI is nowadays considered the most reproducible diagnostic tool for in vivo investigations of excessive trabeculation phenotypes. This paper raises a quite relevant issue, i.e., the current limits in the knowledge of sex-related (besides age-related) normal reference values to be used in the CMR evaluations of myocardial trabeculation. Assessing the normal variability range in different age and sex groups is an important issue in the imaging field for the correct assessment of cardiac diseases, and this topic has been addressed in recent studies (ex: Sci Rep 10:846,2020; Int J Cardiovasc Imaging 2020, 36:2173).

In previous echocardiographic and CMR studies, these volumetric parameters have been shown to be age-dependent, as correctly acknowledged by the Authors in Discussion section. Unfortunately, comparison with previous studies is not always feasible, because of different imaging technologies and/or of different investigation methods (such as including the papillary muscles in the trabeculated muscle mass as in the present paper or in the compact myocardial mass as in other papers). Riffel et al. have also underlined the impact of different strategies on the assessment of the CMR reference values concluding in their CMR study on 362 healthy volunteer that “Quantitative assessment of LV volumes and mass with inclusion of papillary muscles and trabeculae to myocardial mass resulted in significantly different values” (Clin Res Cardiol 2019;108:411). Some of the Authors of the present (under consideration) paper have previously shown that “The TB algorithm could be a consistent method to assess LV and RV CMR values, and to measure trabeculae and papillary muscles quantitatively in various level of experience in CMR (Int J Cardiovasc Imaging 2018 34:1127).

Few data are available as to sex-related differences, corrected by age. The Authors confirmed that males had significantly higher volumes and myocardial mass values and lower EF than females; they also showed that LVCMi and TPMi values (after stratification by age) differed significantly between sexes.

Controversial results have been reported as to possible aging-related changes in LVCMi; anyhow the results of this study regarding LVCMi such as the lack of connection between TrMi and BMI, are confirmed by other studies (see Ref.s # 1,20,32) and/or by anatomic post-mortem investigations that have been conducted in the past (Ref. # 33), respectively.

Although this paper adds a few data to our knowledge of normal populations values for CMR evaluations, its major limit is due to the small number of the subgroups, as stated by the Authors in the final (Limitations of the study) section. Then further studies are required to confirm their (partially novel) data. Such comment might be included in the Conclusions section.

Author’s response

We would like to thank you for the detailed opinion and useful advice. After studying the recommended references, we revised and completed our manuscript, which increased its quality.

We agree with the Reviewer that the small number of subgroups is a limitation that should also be mentioned in the Conclusion section. We added an extra sentence to the final section; please find the corrected version of the Conclusion in the modified manuscript file (page 18, lines 332–335).

Reviewer #2 This paper is interesting in trying to define "normality values" by sex and age. But it fails in the method, in my opinion. It uses the tool that already exists in the market and is known (although very imprecise). The problem is the ventricular volumes that do not coincide with those observed so far (they are too small).

MRI has another tool to assess precision in the quantification of ventricular volumes. We can quantify the flow that leaves the left ventricle in each beat in the aortic root. If they are normal "healthy" patients, this flow must be the same as that obtained by volumes. This part is not included in this paper. If authors want to measure it like that and validate their method they must demonstrate that it is the correct one. I think that is the first thing before defining normality values.

Author’s response

We would like to thank the Reviewer for the comment on the methodological section. The Reviewer is right that the threshold-based (TB) method has been used for years and is commercially available. However, the literature is incomplete in terms of normal values concerning sex and age, particularly when using the TB technique. The advantage of this method is that this technique is highly reproducible and is independent of the experience of the observer [1-3]. Furthermore, it is more accurate than conventional contouring techniques, especially for the determination of trabecular mass. Based on the altering signal intensities, the program automatically identifies each voxel as blood or myocardium, as the trabeculae are counted to the compact mass and extracted from the intracavital volume, resulting in lower volumetric parameters than the conventional contouring technique. This is in line with the Reviewer’s observation.

Riffel et al. observed a strong correlation between conventional and TB methods in measured myocardial mass parameters with the exclusion and inclusion of trabecular and papillary muscles [4]. Another investigation used the TB method of Medis Medical Imaging Systems and compared this semiautomatic segmentation algorithm with the standard method of manual contour tracing for subjects with normal heart function and on a dynamic anthropomorphic heart phantom. The volumetric values calculated in that study were similar to ours, and the correlation between the TB method and manual contouring technique was high in terms of EDV and ESV parameters [2].

We agree with the Reviewer that validation of the software is absolutely necessary for defining normal values. Validation of the TB method was conducted by Varga-Szemes et al., who performed aortic flow analysis to provide independent reference standards for SV measurements, and they described that the software has excellent agreement in SV and CO with the flow reference [1]. Since it has been proven to be a validated method, we did not perform this measurement in our study.

We also made a representative sample from our studied population with 10 people from each subgroup. Please find the presented volumetric, functional and myocardial mass parameters (EDVi, ESVi, SVi, COi, EF, LV_Massi) in Table 1 calculated with conventional technique, with values corresponding to the normal values [5]. The left ventricular mass (LV_Massi) values were calculated as the myocardium between the endo- and epicardial contours and indexed to the body surface area.

IDENTIFIER EDVi (ml/m2) ESVi (ml/m2) SVi (ml/m2) COi (l/m2*min) EF 

(%) LV_Massi (g/m2)

A_1 68.2 28.9 39.2 2.9 57.6 34.2

A_2 93.8 40.6 53.2 3.7 56.7 48.0

A_3 69.8 24.3 45.4 2.9 65.1 38.8

A_4 78.4 29.7 48.8 3.4 62.2 37.5

A_5 93.2 37.6 55.6 5.5 59.7 35.7

A_26 96.0 43.0 53.1 3.5 55.3 49.4

A_27 70.2 27.7 42.5 3.3 60.5 35.9

A_28 86.9 35.9 51.0 3.1 58.7 53.1

A_29 101.2 47.7 53.4 4.8 52.8 50.4

A_30 107.9 42.6 65.3 5.3 60.5 54.9

Females 80.7 32.2 48.4 3.7 60.3 38.8

Males 92.4 39.4 53.1 4.0 57.6 48.7

Average of 

Group A 86.6 35.8 50.8 3.8 58.9 43.8

B_1 87.0 30.9 56.2 3.2 64.5 43.1

B_2 81.0 32.9 48.1 3.7 59.4 37.2

B_3 77.2 32.5 44.7 3.1 57.9 38.5

B_4 88.0 31.7 56.4 3.5 64.0 47.3

B_5 73.2 28.2 45.0 2.8 61.5 38.0

B_26 85.3 34.4 50.9 3.9 59.7 40.8

B_27 109.5 49.4 60.1 4.9 54.9 64.2

B_28 79.3 35.4 43.9 2.4 55.3 42.6

B_29 82.5 36.2 46.4 3.1 56.2 53.7

B_30 94.1 40.7 53.4 3.4 56.8 50.5

Females 81.3 31.2 50.1 3.2 61.5 40.8

Males 90.1 39.2 50.9 3.5 56.6 50.4

Average of 

Group B 85.7 35.2 50.5 3.4 59.1 45.6

C_1 83.4 37.8 45.6 3.7 54.7 43.5

C_3 80.4 29.7 50.7 4.2 63.0 41.4

C_4 73.7 29.2 44.4 2.6 60.3 32.4

C_5 82.3 30.2 52.1 3.9 63.3 39.4

C_6 92.8 37.5 55.2 3.4 59.5 50.3

C_26 91.1 34.0 57.1 3.2 62.7 44.9

C_27 82.9 37.1 45.8 3.0 55.3 48.5

C_28 104.3 43.5 60.8 4.7 58.3 67.9

C_29 88.1 29.0 59.2 4.1 67.1 64.4

C_30 81.2 33.4 47.7 2.2 58.8 43.6

Females 82.5 32.9 49.6 3.6 60.2 41.4

Males 89.5 35.4 54.1 3.4 60.4 53.9

Average of 

Group C 86.0 34.1 51.9 3.5 60.3 47.6

D_1 76.5 25.9 50.6 3.6 66.1 38.1

D_2 76.9 31.6 45.4 2.8 59.0 45.7

D_3 60.6 24.9 35.7 3.3 58.9 37.1

D_4 76.0 23.8 52.2 4.2 68.6 33.3

D_5 76.9 23.5 53.3 3.8 69.4 43.1

D_26 73.5 23.2 50.2 3.0 68.4 42.9

D_27 101.7 39.8 61.9 3.6 60.8 45.3

D_28 66.9 23.8 43.1 2.8 64.4 49.6

D_29 64.1 23.4 40.7 2.3 63.4 42.3

D_30 106.6 39.8 66.8 5.0 62.7 64.7

Females 73.4 25.9 47.4 3.6 64.4 39.5

Males 82.5 30.0 52.5 3.3 63.9 49.0

Average of 

Group D 78.0 28.0 50.0 3.5 64.2 44.2

Table 1 – Representative sample from the studied population divided by age group and sex; five males and five females from each subgroup

EDVi: end-diastolic volume index, ESVi: end-systolic volume index, SVi: stroke volume index, COi: cardiac output index, EF: ejection fraction, LV_Massi: left ventricular end-diastolic myocardial mass index, LV: left ventricular 

Group A: 20–29 years, Group B: 30–39 years, Group C: 40–49 years, Group D: ≥ 50 years

Reviewer #3 The Authors measured the LV mass and volumes in 200 healthy caucasian controls, using a threshold-based (TB) method; in particular, they focussed on the trabeculated and papillary muscle mass (TPM) and on the left ventricular compacted myocardial mass (LVCM), which have been overlooked in previous studies. The article is interesting, even though several points should be addressed.

-As far as patient stratification is concerned, the older group (>50 year old people) might not be representative of elderly people (>60 y.o.), who might display a different phenotype and different reference values than middle-aged patients (50-60 y.o.).

-The threshold-based (TB) method relies on signal thresholding between the myocardium and the bloodpool, so that different thresholds produce different results by including/excluding not only any small trabecula but also any blood/myocardiuam partial volume areas. This is apparent even in Figure 1, where some small trabeculae (not highlighted in green) can be hardly distinguished from the bloodpool. Besides intra- and inter-operator variability, the variability of TB measurements should be tested on cineSSFP images acquired with/without Gd, as well as with different spatial and temporal resolutions.

-Total LV myocardial mass should be measured in systole and diastole, as a double-check, also because this TB method should detect all trabeculae (which are usualy missed with manual countouring) and yield very similar total mass values between sytole/diastole.

-Were all the volunteers free from medical therapy? Please specify in the Methods.

-Table 2 and 3 are more or less the same, with only different subgroup analysis (per sex vs. per age group). I would suggest to merge the two tables. Moreover, I would suggest to represent a column/row about total LV mass (which should be the sum of TPM+LVCM).

Author’s response

Question #1: We agree with the Reviewer that we should have enrolled more elderly people in our study (>60 y.o.). However, in this single-center prospective study, it was quite challenging to include healthy participants older than 60 years, as most of the people in this age group are under medical therapy and have severe comorbidities. As the Reviewer mentioned, elderly people might display different phenotypes than middle-aged people (50–60 y.o.). Although there are investigations reporting normal values for an older population, a different approach was used in these larger center studies [5, 6].

Question #2: The Reviewer’s observation is correct. Visualization of the myocardium (green area in Figure 1) can slightly differ from the calculated myocardium due to the nature of the threshold-based technique: small trabeculae may be calculated as myocardium, even though they are not marked with green color on the image. The threshold was set to default (50%) and was not modified manually in either case; thus, there was no interoperator variability regarding the threshold settings. The Medis Medical System recommends using the software with the default settings, which makes it the most applicable. Other studies have examined the changing threshold to 70% and found that this value might too high [2].

Testing the variability of TB measurements on cineSSFP images acquired with and without Gd is an appropriate question; however, the usage of the contrast agent modified the calculation of the volumes and the masses during the postprocessing evaluation as the border between blood and myocardium became less defined [7]. Because of this, we did not use contrast agent in our study. Please see this text on page 6 line 112–113.

We did not find any data on whether altering spatial and temporal resolution changes the measured TB parameters; however, it would be interesting to compare different settings, which would constitute a completely new investigation. To set the appropriate normal values, we prefer the original setting used in everyday practice in our research.

Question #3: The Reviewer is correct: a double-checking method for systolic and diastolic myocardial mass measurements would be useful mainly in the conventional techniques. However, the TB method calculates the total myocardial mass within the epicardial contour and measures the trabeculae within the endocardial contour automatically with minimal manual correction. Thus, in our study, we did not focus on double-checking because the TB method is an already validated technique with excellent reproducibility [1, 2].

Question #4: Thank you for this comment. We agree that this was not clearly emphasized in the Methods section. All of the participants were free from medical therapy, and information was added to the manuscript. Please see page 4, line 87.

Question #5: The Reviewer requested that we merge Table 2 and Table 3. We hope that the tables are still clear and logical in this form and fit with the Reviewer’s conception. We have also represented the total LV mass values in the table. Please see the new version of it as the new Table 2 on page 9.

1. Varga-Szemes, A., et al., Clinical feasibility of a myocardial signal intensity threshold-based semi-automated cardiac magnetic resonance segmentation method. Eur Radiol, 2016. 26(5): p. 1503-11.

2. Jaspers, K., et al., Improving the reproducibility of MR-derived left ventricular volume and function measurements with a semi-automatic threshold-based segmentation algorithm. Int J Cardiovasc Imaging, 2013. 29(3): p. 617-23.

3. Csecs, I., et al., Left and right ventricular parameters corrected with threshold-based quantification method in a normal cohort analyzed by three independent observers with various training-degree. Int J Cardiovasc Imaging, 2018. 34(7): p. 1127-1133.

4. Riffel, J.H., et al., Cardiovascular magnetic resonance of cardiac morphology and function: impact of different strategies of contour drawing and indexing. Clin Res Cardiol, 2019. 108(4): p. 411-429.

5. Petersen, S.E., et al., Reference ranges for cardiac structure and function using cardiovascular magnetic resonance (CMR) in Caucasians from the UK Biobank population cohort. J Cardiovasc Magn Reson, 2017. 19(1): p. 18.

6. Natori, S., et al., Cardiovascular function in multi-ethnic study of atherosclerosis: normal values by age, sex, and ethnicity. AJR Am J Roentgenol, 2006. 186(6 Suppl 2): p. S357-65.

7. Szucs, A., et al., The effect of contrast agents on left ventricular parameters calculated by a threshold-based software module: does it truly matter? Int J Cardiovasc Imaging, 2019. 35(9): p. 1683-1689.

---

## [Decision Letter · Decision Letter 1]

27 Sep 2021

Sex- and age- specific normal values of left ventricular functional and myocardial mass parameters using threshold-based trabeculae quantification

PONE-D-21-13794R1

Dear Dr. Szűcs,

We’re pleased to inform you that your manuscript has been judged scientifically suitable for publication and will be formally accepted for publication once it meets all outstanding technical requirements.

Kind regards,

Vincenzo Lionetti, M.D., PhD

Academic Editor

PLOS ONE

Additional Editor Comments (optional):

Reviewers' comments:

Reviewer's Responses to Questions

**Comments to the Author**

1. If the authors have adequately addressed your comments raised in a previous round of review and you feel that this manuscript is now acceptable for publication, you may indicate that here to bypass the “Comments to the Author” section, enter your conflict of interest statement in the “Confidential to Editor” section, and submit your "Accept" recommendation.

Reviewer #1: All comments have been addressed

Reviewer #2: All comments have been addressed

Reviewer #3: All comments have been addressed

2. Is the manuscript technically sound, and do the data support the conclusions?

Reviewer #1: Yes

Reviewer #2: Partly

Reviewer #3: Yes

3. Has the statistical analysis been performed appropriately and rigorously? 

Reviewer #1: Yes

Reviewer #2: Yes

Reviewer #3: Yes

4. Have the authors made all data underlying the findings in their manuscript fully available?

Reviewer #1: Yes

Reviewer #2: Yes

Reviewer #3: Yes

5. Is the manuscript presented in an intelligible fashion and written in standard English?

Reviewer #1: Yes

Reviewer #2: Yes

Reviewer #3: Yes

6. Review Comments to the Author

Reviewer #1: The Authors have revised the paper according to the reviewers' comments and the manuscript has been improved by the changes that have been made.

Reviewer #2: As I said in the first review, this paper is interesting in its attempt to define "normality values" by sex and age. But it fails in the method, in my opinion. It uses the tool that already exists on the market and is well known (although very imprecise). The problem is the ventricular volumes that do not match those observed so far (they are too small).

MRI has another tool to assess the accuracy in quantifying ventricular volumes. We can quantify the flow out of the left ventricle at each beat at the aortic root. In normal "healthy" patients, this flow should be the same as that obtained by volumes. This part is not included in this paper and the authors go on to say that it does not validate the method. If the authors want to measure it this way and validate their method, they must prove that it is correct. I think that is the first thing to do before defining normal values.

Reviewer #3: (No Response)

7. PLOS authors have the option to publish the peer review history of their article (what does this mean?). If published, this will include your full peer review and any attached files.

Reviewer #1: No

Reviewer #2: No

Reviewer #3: No

---

## [Editor Report · Acceptance letter]

30 Sep 2021

PONE-D-21-13794R1 

Sex- and age- specific normal values of left ventricular functional and myocardial mass parameters using threshold-based trabeculae quantification 

Dear Dr. Szűcs:

I'm pleased to inform you that your manuscript has been deemed suitable for publication in PLOS ONE. Congratulations! Your manuscript is now with our production department. 

Kind regards, 

on behalf of

Prof. Vincenzo Lionetti 

Academic Editor

PLOS ONE